# In the Shadow of the Pandemic: Examining Therapists’ Perceptions of Work-Related Stress in the Late Stages of the COVID-19 Pandemic in Germany

**DOI:** 10.3390/healthcare12191933

**Published:** 2024-09-26

**Authors:** Smilla Johann, Megan Evans, Rike Böttcher, Nadine Muller, Barbara Buchberger, Charbel El Bcheraoui, Heide Weishaar

**Affiliations:** 1Evidence-Based Public Health Unit, Centre for International Health Protection, Robert Koch Institute, 13353 Berlin, Germany; smillajhn@gmail.com (S.J.); megan.evans45@nhs.net (M.E.); r.m.boettcher@gmx.de (R.B.); buchbergerb@rki.de (B.B.); el-bcheraouic@rki.de (C.E.B.); 2Speciality Network Infectious Diseases and Respiratory Medicine, Charité-Universitätsmedizin Berlin, 13353 Berlin, Germany; nadine.muller@charite.de

**Keywords:** occupational stress, COVID-19, therapists, healthcare, health personnel, working conditions

## Abstract

Background/Objectives: The previous literature has identified increased work-related stress among healthcare workers (HCWs) during the COVID-19 pandemic. This study analyzes work-related perceived stress experienced by therapists during the COVID-19 pandemic in Germany to identify potential for supporting this crucial group of HCWs in future health crises. Methods: Survey data on stress, measured through the Perceived Stress Scale 4, among HCWs (therapists n = 612, nurses n = 501, and doctors n = 461) were analyzed using descriptive statistics, and data from four semi-structured interviews and seven focus group discussions with therapists were analyzed using thematic content analysis. Data were collected from March to September 2022. Results: Therapists reported similar stress levels to other HCWs, with the reporting of perceived stress differing between work contexts. Eight stressors were identified through the thematic analysis: (1) concerns about maintaining the quality of care, (2) uncertainty about the future, (3) workload, (4) interactions with colleagues and patients, (5) PPE, (6) the risk of infection, (7) insufficient information flow, and (8) the lack of public and political recognition. Conclusions: This study emphasizes the distinct challenges that therapists experienced during the COVID-19 pandemic. By identifying the factors that contributed to the stress experienced, the study can inform targeted support strategies which can enhance therapists’ work, ultimately contributing to sustaining essential healthcare services during public health crises.

## 1. Introduction

The COVID-19 pandemic has been a unique challenge to the resilience of health systems worldwide, exposing deficits and vulnerabilities. While a broad range of individuals [1] and professions [2,3] were negatively affected by the pandemic, healthcare workers (HCWs) were among the groups mostly suffering from the consequences of dysfunctional health systems, evidenced by the increased levels of stress and mental health issues [4,5,6,7,8,9,10]. The World Health Organization (WHO) defines stress as “a state of worry or mental tension caused by a difficult situation” [11]. The previous literature examining HCWs’ experiences during the COVID-19 pandemic suggests that stress resulted from factors such as working with personal protective equipment (PPE) [12,13,14], increased workload [10,12,13,15,16], a lack of knowledge about the virus and uncertainty about the future, [13,17] paired with financial insecurity [18]. Previous research emphasizes challenges in collegial relationships due to social distancing [19] and difficulties in communication with patients [20,21,22,23,24] as a result of working with PPE and reduced face-to-face visits or the switch to telehealth [25,26,27]. Studies further highlight that a lack of institutional communication and coordination [16,28,29,30], limited opportunities for participatory decision-making [31,32], limited psychological, managerial, material, and technical support [10,33,34,35], and dissatisfaction with governmental responses [36,37] contributed to stress among HCWs. Work-related stressors seemed to differ depending on the place of work [12,13], with Frenkel and colleagues highlighting greater stress among HCWs in the outpatient compared to the hospital sector [13]. While most studies focus on the stress experienced by the hospital staff with direct contact with COVID-19 patients, notably nurses, doctors, and trainees [17,26,38], other HCWs, such as therapists, have received little attention from researchers, and comparative analyses of profession-specific challenges are lacking [39].

Drawing on the German professional group of “Heilmittelerbringer*innen”, this article focuses on therapists as HCWs providing treatments aimed at preventing, curing, or alleviating post-treatment complaints [40,41]. The specific therapeutic professions included in this group are physiotherapists, speech, language, and swallowing therapists (in short: SLTs), occupational therapists, podiatric therapists, and nutritional therapists [40]. Therapists play a crucial role in the healthcare system and are involved in diverse clinical care scenarios [42], especially in the care of chronically ill patients, often providing a key link between acute care and rehabilitation [43].

This study provides comprehensive insights into therapists’ perceptions of work-related stress during the COVID-19 pandemic in Germany, thereby providing the evidence base that is needed to develop tailored strategies to support this crucial professional group in future public health crises.

## 2. Materials and Methods

This study was embedded within a larger mixed-method study investigating the experiences of HCWs during the COVID-19 pandemic in Germany and several African countries [44]. We combined data from a quantitative survey, semi-structured interviews (SSIs), and focus group discussions (FGDs); the analysis that led to this paper focused on the German arm of the study. This manuscript jointly presents the findings from the quantitative and the qualitative part of the study conducted in Germany. While the two parts were conducted independently and in parallel, the data were compared and triangulated at the analysis stage in order to provide a comprehensive account of the therapists’ experiences with stress during the COVID-19 pandemic.

Sampling for the quantitative survey was conducted in two stages. First, 3 of 16 federal states in Germany were selected as study regions using a stratified random sample. Three strata (low, medium, and high COVID-19 burden) were defined based on the cumulative number of COVID-19-related deaths per 100,000 inhabitants since the beginning of the pandemic, as reported by the national COVID-19 reporting system [45]. Five to six federal states were assigned to each stratum before a random sample was drawn from each stratum. Second, cluster sampling was used to secure a representative sample of healthcare facilities spanning all tiers of the healthcare system, with the following categories of healthcare facilities serving as clusters: primary and standard care hospitals; maximum care hospitals; long-term care facilities; outpatient care providers; and general practitioner practices. In March 2022, all employees from the sampled healthcare facilities were invited to take part in an online survey if they were aged 18 or above and employed in a healthcare facility during the pandemic. Due to the initial suboptimal response rates, the recruitment approach was adjusted in May 2022 by approaching professional associations as gatekeepers and allowed HCWs from all regions of Germany to participate in the study.

The survey data were collected using an online questionnaire hosted on the online platform VOXCO between March and October 2022. The questionnaire was developed based on insights obtained from a scoping review [39]. It contained the four-item Perceived Stress Scale (PSS-4) [46] and a question on which of a selected number of factors was perceived as the most important in making the respondent’s life more difficult or more stressful since the beginning of the pandemic. The questionnaire underwent piloting through cognitive interviews with four HCWs to ensure clarity and gauge time commitment. In addition to questions about perceived stress, the questionnaire included questions on sociodemographics, and several questions to explore factors that had been identified in a previous literature review [39] as being associated with stress, including questions on concerns and worries, the work environment, and on perceived social and institutional and governmental support. A descriptive analysis of perceived work-related stress and these factors, comparing therapists with doctors and nurses, was carried out in order to provide a brief comparison of stress levels reported by therapists and other healthcare professionals and a first insight into the factors that were associated with self-reported stress among the different professional groups.

The qualitative part of the study served to gain an in-depth understanding of therapists’ experiences with stress and the factors that were perceived to contribute to stress. Initially, participants for the SSIs and FGDs were recruited from the survey participants. Upon completion of the survey, respondents had the option to volunteer to take part in the qualitative part of the study. Volunteers were then approached with a request for participation in an SSI or FGD. After adjustment of the recruitment strategy, professional associations and networks were asked to disseminate the call for participation in SSIs and FGDs among their members and via their communication channels. Selection of interviewees and FGD participants was done in a way to ensure a diverse representation of HCWs from various regions, areas of work, genders, and professions aiming to capture a variety of views. All SSIs and FGDs were conducted between April and December 2022, online, in German, and were audio-recorded. In addition to one lead interviewer, the data collection was supported by a second researcher who assisted with documentation, recording, and logistical matters. The topic guides for the SSIs and FGDs were developed using insights obtained from a scoping review on HCWs’ experiences during the COVID-19 pandemic [39]. Key themes included the following: HCWs’ perceptions of the pandemic situation; challenges; factors influencing HCWs’ experiences and work situations; support needs; and areas of potential improvement. SSIs lasted on average 55 min and FGDs lasted 70 min. Data collection was stopped as data saturation was reached. The transcripts of SSIs and FGDs were analyzed using thematic content analysis based on a category system [47]. Codes were first developed deductively drawing on Karasek’s Job Demand-Control Model [48] and Siegrist’s effort reward imbalance model [49]. The codebook was subsequently expanded using an inductive approach.

Throughout data collection, regular data quality checks were conducted. The integration of the quantitative and qualitative data was based on the in-depth model according to Mayring [50]. In a first step, the survey data were analyzed using descriptive statistics to provide initial findings on the stress levels reported by therapists in relation to other HCWs and associated factors. Subsequently, a thematic analysis of the SSI and FGD data was undertaken to develop an in-depth understanding of the factors that were reported as contributing to perceived stress among therapists.

All potential study participants received written information about the study before being asked to participate. Participation in the study was voluntary and based on written consent. Ethical approval was obtained from the Charité-Universitätsmedizin Berlin, Germany. Data protection approval was acquired from the data protection officer of the RKI. The study was funded by the German Federal Ministry of Health through the Global Health Protection Programme (Kapitel 1505 Titel 68601). The funders played no role in the design, conduct, or reporting of this study.

## 3. Results

### 3.1. Description of the Sample

The survey sample comprised 1574 HCWs, including therapists (n = 612), nurses (n = 501), and doctors (n = 461). Table 1 displays the sociodemographic data of survey respondents.

Fifteen therapists enrolled in the qualitative part of the study and completed an SSI (n = 4) or participated in a FGD (n = 11). The sample consisted of 12 physiotherapists and three SLTs. Four SSIs and seven FGDs were included in the analysis. The sociodemographic data of interviewees and FGD participants are displayed in Table 2.

### 3.2. Work-Related Stress Experienced by Therapists in Comparison to Doctors and Nurses

The survey data show that different professional groups reported similar stress levels. Overall, the median PSS-4 score was 13, ranging from 4 to 20. The median score was also 13 for each of the three categories of HCWs, ranging between 7 and 19 for doctors, 5 and 18 for nurses, and 4 and 20 for therapists. The interquartile range of the PSS-4 score was two overall as well as for the three categories of HCWs. However, perceived stress was associated with different factors in the different professional groups (Figure 1). Working with PPE (28.2%) appeared to be the primary stressor for therapists (compared to workload among doctors and nurses, 50.2% and 39.3%, respectively), followed by the fear of infection (20.9%), a lack of social gatherings (20.1%), and workload (19.3%). In comparison, doctors and nurses showed a lower level of concerns regarding the fear of infection (doctors: 12.2%, nurses: 13.0%) and working with PPE (doctors: 6.1%, nurses: 14.8%).

### 3.3. Categories of Stressors Mentioned by Therapists

To gain more detailed insights into the stress experienced by therapists specifically and the factors contributing to their perceived stress, the eight themes that were identified through the thematic analysis of the qualitative data are outlined below. They are listed in order of ascending prominence in the data, with the participants’ quotes being provided as illustrative examples to highlight the specific aspects of each theme. Depending on the preferences of the respective interviewee or FGD participant, each quote is allocated to the participant’s personal data, or only generic information is given that the quote was provided by a study participant.

#### 3.3.1. Concerns about Maintaining Quality of Care

Therapists described concerns and challenges related to maintaining good quality health services during the COVID-19 pandemic. They reported that maintaining good quality of care was difficult because of physical distancing and extended protective and hygiene measures, such as the requirement for both patients and therapists to wear masks during speech and language therapy or for keeping the mandated 1.5 m safety distance during therapeutic treatment. Therapists highlighted the necessity of close patient contact, which made it impractical to adhere to COVID-19-related guidelines and regulations. Additionally, it was reported that patients frequently canceled appointments and staff were absent due to illness, adversely affecting therapy regularity and success.

#### 3.3.2. Uncertainty about the Future

Participants worried about their professional future and their ability to sustain their economic basis, with financial concerns being a particularly prominent theme among self-employed therapists and practice owners who reported major losses in turnover in the early months and peak phases of the pandemic. Patients canceling appointments, combined with the initial lack of financial support for therapists from the state, contributed to financial bottlenecks. Highlighting that government aid packages were not initially directed towards the therapeutic sector, an SLT reported the following:

*“Especially when there were the waves, the different waves, and the patients canceled. And simply no patients came. Then there was a time when we weren’t allowed to do video therapy. So, it was a time of absolute financial uncertainty.”* (SLT, FGD).

Practice owners also highlighted the significant expenses for acquiring PPE. The state’s support in the form of a “hygiene flat rate” (SLT, FGD) was reported to be insufficient in covering the actual costs for protective masks and disinfectants, with an interviewee stating that he was on the “verge of insolvency” (physiotherapist, FGD) as a result. Financial concerns also extended to employed therapists, with interviewees mentioning short-time work due to the decline in treatments and job insecurity due to not complying with the legally required COVID-19 vaccination for HCWs.

#### 3.3.3. Workload

In contrast, other therapists attributed the high stress they experienced to an increase in working hours or additional working requirements. It seemed that particularly, therapists working in hospitals faced a surge in workload during the pandemic, partly due to increased sick leave among colleagues due to COVID-19, quarantine restrictions, exhaustion, and colleagues needing “a break” (physiotherapist, FGD) from perceived pandemic stress. Practice owners recalled an increased workload due to the need to inform themselves and the bureaucratic workload resulting from COVID-19-related regulations (including the procurement of PPE and testing requirements). Increased bureaucracy led to less time for the actual treatment of patients, as reported by a practice owner:

*“I have to do it all myself and that’s working time. It’s a burden with the working time, a workload that is then no longer left for my patients.”* (physiotherapist, FGD).

Two interviewees reported that they were considering closing their practice because of their frustration with the enormous bureaucratic effort combined with the pre-existing low remuneration for treatment.

#### 3.3.4. Interactions with Colleagues and Patients

While positive interactions in the workplace were perceived as a supportive factor by some therapists, several participants also described interactions with patients and colleagues as a potential source of perceived stress. They reported tensions within the team and challenging discussions with patients to have a negative impact on the social climate in the workplace. Interviewees and FGD participants reported that, due to the fear of infection, lockdown, time constraints, and increased workload, colleagues and participants were often in a foul mood and tense, increasingly focused on their own interests, and quick to feel attacked, leading to an increased potential for communication escalation: 

*“It’s relatively clear that communication has become much more difficult for whatever reason. We’re actually seeing that people are getting agitated very quickly, feeling attacked or overloaded for all kinds of reasons.”* (physiotherapist, FGD).

Further, patients’ anxieties and a lack of understanding for, and adherence to, COVID-19 protocols was mentioned as making communication challenging and as “exhausting […and…] tiring” (study participant, SSI). The wearing of face masks seemed to be a particularly frequent issue of contention, with instances where therapists had to ask patients to leave the practice due to not complying with hygiene measures:

*“We really got to know patients from a different side, where we actually kicked patients out of the practice.”* (physiotherapist, FGD).

#### 3.3.5. PPE

Therapists frequently reported stress resulting from working with and procuring PPE during the COVID-19 pandemic. Particularly, those who were not used to wearing PPE reported that working with PPE was physically demanding and noted negative impacts on their health, including symptoms like headaches and fatigue:

*“What I still find really difficult is working with FFP2, I honestly admit, because we can’t take breaks during therapy. We simply work to a tight schedule and we can’t take the prescribed break in between. I notice that at the end of the day, I just have a headache.”* (physiotherapist, FGD).

Another burden in relation to PPE was procurement. While those working in hospitals were less likely to report PPE shortages, self-employed therapists and therapists working in private practices encountered procurement challenges. Some reported being highly proactive in addressing these challenges, with instances of therapists sewing masks and personally securing PPE through local pharmacies.

#### 3.3.6. Risk of Infection

Many participants reported feelings of stress resulting from the fear of contracting COVID-19 in the workplace. These participants not only conveyed the perception of their own risk of infection, but also frequently expressed a fear of infecting others. In particular, therapists reported concerns about transmitting the virus to patients, many of which belonged to high-risk groups and were thus, more likely to contract and suffer from the consequences of an infection with SARS-CoV-2:

*“It was really the case that we were actually scared. For ourselves. But also, for the patients. Most of the staff were also concerned about protecting others and we were really very careful about that.”* (physiotherapist, SSI).

Particularly SLTs who applied intraoral therapy methods noted that it was difficult for them to weigh up the risk of infection and the need to use PPE against the risk and negative consequences of ineffective therapy:

*“There is always the issue of risk assessment. Once these regulations come in that say 1.5 m distance, FFP2 mask,… And then look: How risky is my work! And it is absolutely risky. And I have to get close to the mouth of a swallowing patient, which means I can’t avoid gowns and goggles at all. Or at least at the beginning. And I couldn’t get around the mask at all.”* (SLT, FGD).

Therapists further reported that they were often unable to adequately protect themselves from infection in their professional practice due to not having access to medical face masks at the beginning of the pandemic and, therefore, coming up with makeshift solutions. Only two physiotherapists held contrasting views, stating that they had not been concerned about infection as they considered COVID-19 to be a “virus like any other” (physiotherapist, FGD).

#### 3.3.7. Insufficient Information Flow

Almost all participants reported a lack of accessibility, clarity, and dissemination of the information necessary for therapists to practice their profession. This lack of profession-specific information regarding COVID-19 regulations had negative implications for the participants’ professional practice.

*“There was no reference book or website where I could look up how the infection was going and what protective measures I had [to implement].”* (study participant, SSI).

Most therapists described not only difficulties in gaining access to relevant information, but also reported that the available information was often unclear and contradictory. They noted receiving contradictory information from local or federal authorities and professional associations, attributing this to disagreement and unclear responsibilities between health authorities. The difficulty to identify reliable and up-to-date information was perceived as a particular problem as therapists often took the role of mediators of the pandemic and regulatory information and, thus, were often overwhelmed by the task of informing patients. Practice owners highlighted particular challenges in accessing information, with many spending considerable time on searching and reviewing information, which consumed valuable time. Clinical therapists, on the other hand, seemed less likely to experience insufficient information flow. While the lack of profession-specific information was also evident here, clinical therapists seemed to be more likely to receive information from hospital management.

#### 3.3.8. Lack of Political and Public Recognition

Therapists frequently reported a perceived lack of recognition by state institutions, the media, and the general public, and highlighted that this put an additional burden on them. Therapists described that they did not feel recognized in their role as members of a medical, system-relevant profession. Some reported that particularly early in the pandemic, physiotherapy practices were treated similarly to massage or tattoo studios regarding COVID-19 measures, with regulations requiring the closure of practices, causing frustration on part of the practice owners and employees:

*“So we started in the early days of the pandemic, when we were banned from practicing our profession, which hit us hard. [...] And I thought: But we’re still involved in patient care.”* (SLT, FGD).

The lack of recognition was further perceived to be evident in insufficient government support, both financially and in terms of obtaining PPE and profession-specific information. For many participants, this led to the impression that they had been “forgotten throughout the pandemic” (physiotherapist, FGD). Participants contrasted the initial failure of the government to provide financial aid and bonus payments for therapists with the financial support that other HCWs and so-called system-relevant professions had been promised to receive. Participants reported that they felt “at the bottom of the list of needs” (SLT, FGD) and less prioritized compared to HCWs in other sectors. They highlighted discrepancies between the moral obligation that had been posed on them and the responsibility they had felt with regard to having to maintain patient care. They recalled feeling “left alone” (physiotherapist, FGD) and having to find individual, often provisional and imperfect, solutions to abiding by regulations and protecting themselves and patients from being infected.

*“If I say to several professional groups: ‘You are now systemically relevant, you are part of a task force, you have to work in this pandemic!’ Then I have to take very, very good care of them. And really take care of them and not just let them fall by the wayside and say: ‘Improvise!’”* (physiotherapist, FGD).

Some interviewees suggested that the lack of recognition during the pandemic was an aggravation of the pre-existing societal and political neglect of the importance of therapists and their contribution to patient care.

## 4. Discussion

This mixed-methods study investigates therapists’ perceptions of work-related self-reported stress during the COVID-19 pandemic in Germany, focusing on a professional group overlooked in academic and political discourse. The quantitative analysis, which contrasts perceived stress levels and factors associated with perceived stress reported by therapists, doctors, and nurses, provides a brief descriptive overview of perceived stress levels and the relative importance of different pre-identified factors associated with perceived stress among different HCWs. The qualitative, thematic analysis offers an in-depth analysis of the factors that were raised by therapists in SSIs and FGDs as contributing to work-related self-reported stress. To our knowledge, our study is the first to investigate therapists’ experiences of work-related self-reported stress during the COVID-19 pandemic. It thus, puts the spotlight on an under-researched professional group which plays a crucial role in the healthcare system and in providing care to the most vulnerable patients. The analysis presented above not only contributes to a better understanding of the situation of therapists and draws attention to profession-specific challenges during a public health crisis, but also provides the evidence-base that can inform the development and implementation of tailored support for a key group of HCWs in pandemic and non-pandemic times.

The survey data analysis shows that perceived stress, measured using the international standardized instrument PSS-4, did not differ between therapists, doctors, and nurses. Yet, differences exist between the three professional groups with regard to the factors that are associated with, and perceived to contribute to, self-reported stress. The quantitative as well as the qualitative analysis shows that for therapists, having to work with PPE and the risk of a COVID-19 infection were important sources of work-related self-reported stress. Regarding PPE, perceived stress was caused by the actual physical impairments derived from working with PPE, the difficulty to provide adequate patient care when having to use PPE, and problems relating to PPE availability and the supply chain, especially for independent therapy practices. The findings are in line with the existing literature, identifying the use of PPE as a key stressor during the pandemic [51,52,53]. Our analysis further suggests that the risk of a COVID-19 infection, when coupled with a lack of PPE, contributed to a high perceived professional risk and a low ability to control work-related risks. The literature suggests that alternatives were sought, with some SLTs changing to teletherapy [54,55]. However, our study highlights the inherent limitations of such alternatives due to the perceived inferior value of telehealth among therapists and patients as well as a lack of physical interaction and mastery of technology [56,57].

It needs to be acknowledged that perceptions of stress differ between individuals and according to contextual factors and can change depending on the environment of an individual. Further, the potential impact that a stressor has on an individual’s emotional or mental health is mediated by various factors, including the individual’s cognitive appraisal of the situation and the internal and external resources they can mobilize [58]. In this manuscript, we only focus on perceived stress and the factors that were perceived as contributing to stress. We do not address other factors like appraisal or health outcomes in any detail in this manuscript. For example, the data do not allow us to draw conclusions about the actual psychological, physical, or other consequences of the perceived stress nor to analyze any differences in the stress’s impact on different professional groups. While the actual health implications of the perceived stress were not measured and can, therefore, not be quantified, well-established theories on the correlation between perceived stress and negative health outcomes [58] suggest that perceived stress might have a negative impact on HCW health.

Given that appraisal as a mediating factor has an impact on the perceptions of stress, any comparisons between different groups or environments come with limitations. Our combined analysis, however, suggests that therapists who worked in certain facilities and settings often mentioned similar factors as contributing to their stress experiences, thereby pointing to context-specific challenges. For example, staff who worked in the ambulatory sector, including in (often small) therapy practices, seemed to be particularly affected by the lack of PPE and insufficient information flows due to a number of reasons. Not being part of a large organization, whose management provided information and guidance for all staff meant that therapists carried considerable individual responsibility for implementing hygiene measures. Lacking information and profession-specific guidance about how to implement regulations while protecting themselves and patients from infection and maintaining quality of care was perceived as stressful [52,59,60]. Self-employed therapists, practice owners in particular, further saw themselves confronted with work-related stress due to the additional bureaucratic workload, financial worries, and job insecurity. Here, pandemic-specific challenges seemed to be paired with already limited profit margins in therapeutic practices [61]. Existing studies underpin and provide some background information to the severity of these financial concerns: according to an analysis of the financial situation in ambulatory physiotherapy practices during the first wave of the COVID-19 pandemic, 45% of all physiotherapy practices experienced a decrease in utilization to less than half, with 15% temporarily closing, and 58% reporting implementing short-time work [62].

The research to date indicates that HCWs in the outpatient sector are particularly affected by a lack of public and state appreciation [26,29,37]. Our study supports this research by adding the perspective of employees and owners of therapeutic practices. These respondents had particularly strong feelings of being neglected and forgotten during the pandemic. Given that this study indicates that therapists working in inpatient and outpatient care experienced different stressful factors during the pandemic, future studies should compare the experiences of therapists in different types of health facilities to identify variations and specific challenges. This is even more important as more than half of all physiotherapists in Germany work in private practices (55.6%), with a much smaller proportion of physiotherapists working in hospitals (27.8%) and care facilities (16.7%) [63]. Also, private therapeutic practices play a pivotal role in delivering essential health services especially in rural areas and home settings as well as to chronically ill and elderly patients and thus, in ensuring service provision to those who require it most [64]. Supporting therapists in fulfilling these roles in pandemic as well as non-pandemic times can help to maintain patient care, prevent high rates of hospital admission of vulnerable patient groups, and relieve the inpatient sector from additional work.

Our analysis suggests that therapists have a strong sense of responsibility for providing good quality health care and maintaining health services [65]. During the COVID-19 pandemic, this sense of responsibility seemed to be, on the one hand, fueled by the governmental and public expectations of vis-à-vis system-relevant professions, and on the other hand, undermined by the lack of governmental support and recognition of therapists as an important system-relevant group. According to Siegrist’s effort reward imbalance model, feelings of work-related stress increase due to an imbalance between work performance and recognition, leading to so-called gratification crises [49]. Similar processes and the risk of a resulting gratification crisis can be assumed from the therapists’ perceptions of being undervalued and not recognized as a crucial system-relevant profession due to the lack of governmental support during the pandemic.

While this paper focuses on the therapists’ work-related self-reported stress during the COVID-19 pandemic, the analysis supports the other literature that highlights the pre-existing factors that contribute to work-related self-reported stress and frustration among therapists, which were exacerbated by the pandemic [62]. Pre-existing deficits include financial challenges, a lack of economic viability, high levels of bureaucracy, a lack of public and political recognition of therapists as important providers of health services, and staff shortages. In fact, data on physiotherapists in Germany alarmingly show that skilled therapists are missing in every federal state (apart from Hamburg) and that staff shortages are growing [66,67]. For many years, therapists have advocated for their professions to be publicly and politically more recognized [68,69]. The demographic change and rising healthcare demand for chronically ill and elderly patients increases the urgency of the increased recognition of therapists and the development of sustainable solutions for the growing staff shortages in the therapeutic sector [68,70]. Advancing the academization of therapeutic professions and aligning their education with practices in other European countries [71], as has long been requested by professional associations [69,72], could be a first crucial step to achieving this aim [70]. Another suggestion to improve the recognition and attractiveness of therapeutic professions and ensuring financial security is restructuring performance compensation by incorporating organizational efforts [61].

This study has a number of limitations. First, the views presented in this paper are those of the individual study participants, with a self-selection bias. A large majority of the study participants were female and between 41 and 60 years old, with many having extensive professional experience. Due to the unavailability of reliable sociodemographic data, no comparison encompassing all therapeutic professions in Germany can be made. Existing sociodemographic data on physiotherapists show that physiotherapy is a female-dominated and young profession [73,74]. Given that sociodemographic characteristics are likely to influence HCWs’ experiences in a pandemic situation [75], the higher proportion of older participants compared to the actual demographic distribution in the professions investigated might have influenced the analysis. A second limitation is that only physiotherapists and SLTs participated in the SSIs and FGDs and we could not differentiate between different therapeutic professions in the quantitative analysis. This limitation is minimized by the fact that physiotherapists constitute the largest group of therapists [42] and by the fact that there is no reason to assume that the work-related stress experienced by occupational therapists, podiatric therapists, and nutritional therapists differs strongly from that of physiotherapists and SLTs. However, future studies should comparatively investigate the different therapeutic professions to analyze profession-specific challenges and support needs as well as potential differences. Third, the presented statistical analysis provides a brief description of the work-related self-reported stress of therapists, nurses, and doctors and associated factors but lacks inferential statistical predictions.

## 5. Conclusions

This study underscores the importance of supporting therapists in times of pandemics and public health crises and identifies potential areas for improvement. Our analysis demonstrates unequivocally that strategies, including political interventions, to improve the working conditions of therapists in Germany are urgently needed to strengthen the health system and ensure the provision of crucial health services. First and foremost, it needs to be ensured that therapists receive the material and technical support, including PPE, test equipment, and financial assistance, to allow them to provide continuous patient care during pandemics. In addition, improved communication and profession-specific information should be made available in such times to support therapists in fulfilling their crucial role as health service providers, particularly with regard to vulnerable patient groups. As indicated in prior research, professional training [62], (preventive) psychological support [51,76], and advanced communication strategies [18] can function as supportive elements. Arguably more importantly, pre-existing deficits related to the working conditions, remuneration, and education of therapists have to be tackled to combat the shortage of skilled professionals and ensure that sufficient human, financial, and technical resources exist to guarantee good quality patient care. Particular attention should be paid to supporting therapists in the outpatient sector.

## Figures and Tables

**Figure 1 healthcare-12-01933-f001:**
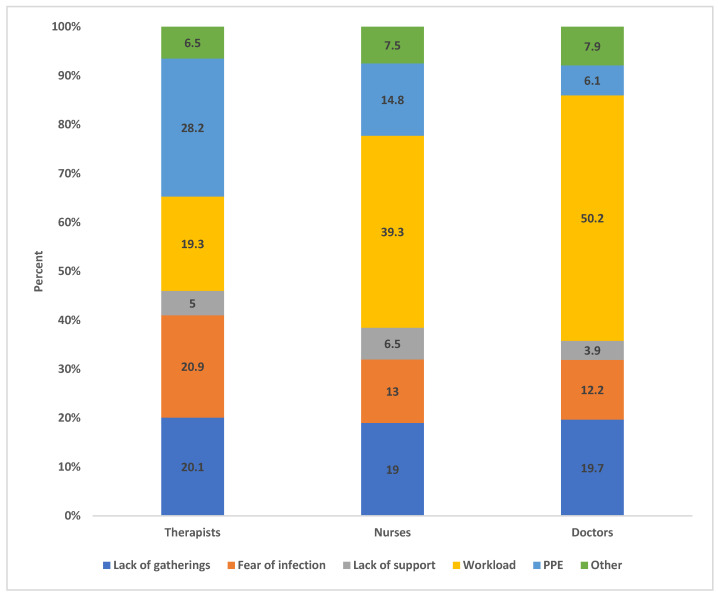
Comparison of factors associated with self-reported stress by professional group.

**Table 1 healthcare-12-01933-t001:** Sample description of survey respondents analyzing HCWs’ reports of stress during the COVID-19 pandemic in Germany, 2022.

	Therapists	Nurses	Doctors	Total
	n = 612	n = 501	n = 461	n = 1574
	No. (%)	No. (%)	No. (%)	No. (%)
Gender				
Female	473 (77.3)	397 (79.2)	274 (59.4)	1144 (72.6)
Male	133 (21.7)	93 (18.6)	178 (38.6)	404 (25.7)
Not specified	6 (1.0)	11 (2.2)	9 (2.0)	26 (1.7)
Age (in years)				
30 or younger	55 (8.9)	95 (19.0)	20 (4.3)	170 (10.8)
31–40	161 (26.3)	108 (21.6)	92 (20.0)	361 (22.9)
41–50	170 (27.8)	103 (20.6)	131 (28.4)	404 (25.7)
51–60	168 (27.5)	132 (26.3)	141 (30.6)	441 (28.0)
60 or older	35 (5.7)	25 (5.0)	62 (13.5)	122 (7.7)
Not specified	23 (3.8)	38 (7.6)	15 (3.3)	76 (4.8)
Professional experience (in years)				
10 or less	101 (16.5)	117 (23.4)	94 (20.4)	312 (19.8)
11–20	196 (32.0)	118 (23.6)	142 (30.8)	456 (29.0)
21 or more	306 (50.0)	236 (47.1)	218 (47.3)	760 (48.3)
Not specified	9 (1.5)	30 (6.0)	7 (1.5)	46 (2.9)

**Table 2 healthcare-12-01933-t002:** Sociodemographics of interviewees and FGD participants investigating therapists’ perceptions of stress and factors contributing to perceived stress during the COVID-19 pandemic in Germany, 2022.

	n = 15
	No. (%)
Gender	
Female	11 (73.3)
Male	4 (26.7)
Not specified	0 (0.0)
Age (in years)	
30 or younger	1 (6.7)
31–40	2 (13.3)
41–50	3 (20.0)
51–60	8 (53.3)
60 or older	1 (6.7)
Not specified	0 (0.0)

## Data Availability

The datasets generated for this study are available on request to the corresponding author.

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
