# Peer review of "In the Shadow of the Pandemic: Examining Therapists’ Perceptions of Work-Related Stress in the Late Stages of the COVID-19 Pandemic in Germany"

_healthcare, 2024, doi:10.3390/healthcare12191933_

Round 1
Reviewer 1 Report (Previous Reviewer 1)
Comments and Suggestions for Authors
The manuscript's evaluation is as follows:
The introduction and research hypotheses section is enriched with the latest references, showcasing the novelty of the study. The findings are well-stated and effectively address my inquiries. However, in the methods section, the Confirmatory Factor Analysis pertaining to concerns such as maintaining quality of care, uncertainty about the future, workload, interactions with colleagues and patients, PPE, risk of infection, insufficient information flow, and lack of public and political recognition should be reported with current data. I am more than happy with this, as the quality of the instrument reflects the quality of the research. It is imperative that reliability and validity are clearly reported.
Author Response
Please see the attachment

Reviewer 2 Report (New Reviewer)
Comments and Suggestions for Authors
I think it is an interesting topic that makes visible a group of workers and their fear and tension in the face of the pandemic. However, there are several points that would be worth highlighting and/or modifying to better locate the results.
I think that in the introduction, what stress is should be defined, perhaps based on the definition of the World Health Organization. It should be clarified that it is a different definition than that of anxiety, depression and other mental disorders, although they are probably associated.
In summary and in other sections of the text, they talk about “stress levels”, for example in summary: “Therapists reported similar stress-levels to other HCW, with reporting of stress differing between work context.” In the instrument they used, based on a previous study (https://doi.org/10.1371/journal.pone.0254252), it never deals with “stress levels”, nor does it quantify or validate such levels. For the above, I consider that the present study cannot mention “stress levels”. This is handled in the summary and in other parts of the text. I consider the study to be valuable and provides relevant information, without the need to manage these “stress levels”, which are also not mentioned in the text. What can be mentioned and is clearly referred to in the results are the stressors, which are different between the groups studied.
In the figure, the text within the figure is blurry and small, please improve.
In the results it should always be mentioned as “perceived stress”
In the discussion it should be clarified that perceived stress is a broad term and that by itself it does not imply a mental alteration. Perception is different for each person and depends on what you do and what surrounds you and your background. The above, so that the reader does not confuse that having stress implies a mental alteration. Perceived stress can change depending on the environment in which you are. Each profession had different stressors during the pandemic, but care must be taken to understand that stress is a psychological construction of each individual, and although two people manifest stress under the same definition, perhaps the factors and health consequences may be different, so comparing them is complex. For example, a doctor or nurse in a COVID unit who saw many people die every day, despite working hard and with extreme fatigue, can be said to have been stressed and it brought consequences. Another professional may also say that she is stressed, but the psychological consequences she suffers from it may be different. It is a complex issue that the authors should try to manage so as not to underestimate the stress that a person feels for whatever reason, since it is a real perception, but not try to compare them, since they can be in different environments and with different health consequences.
The topics discussed about therapists' perceptions seem relevant to me and are important to consider in future emergencies
In discussions at some point it must be stated that we could all suffer from stress, whether health workers, but also workers in other areas, students, etc. and for different reasons. For example, it highlights the economic difficulties and public policies that did not attend equally to all sectors. I continue to give the example of other forgotten sectors, such as street vendors, or informal or temporary workers, who were left unprotected, mentioning that it was a problem in different types of workers, not only in some sectors of health workers (https://pubmed.ncbi.nlm.nih.gov/34898485/)
Round 2
Reviewer 2 Report (New Reviewer)
Comments and Suggestions for Authors
The manuscript was significantly improved. However, there is still one aspect that needs to be addressed in more detail.
In the previous comment, the observation was made about “stress levels.” Now they have made clear the instrument they used, which was the Perceived Stress Scale (PSS-4). By maintaining in the summary and other parts of the text a comparison between “stress levels,” it is necessary to add a comparison of the scores of said instrument in the different analysis groups, making sure to use the measures of central tendency, dispersion and statistical methods appropriate for said variable. The above in order to be able to compare the “stress levels” between the groups. Also add in the discussions what is pertinent to this result. I suggest adding in the discussions that although the stress level is the same between the different groups, it is not implicit that the psychological consequences are the same between them, an aspect that they already mentioned that was not analyzed.
In the summary it is mentioned that “Therapists reported similar stress-levels to other HCW.” I consider it relevant to mention what the main stressor was in each group, because although the level may be the same, what stressed the different groups was very different.
Author Response
Please see the attachment

This manuscript is a resubmission of an earlier submission. The following is a list of the peer review reports and author responses from that submission.
Round 1
Reviewer 1 Report
Comments and Suggestions for Authors
Review for Author(s)
General comments · I would like to begin by expressing my gratitude to the Author(s) for providing me with the opportunity to review the scientific work titled "In the Shadow of the Pandemic: Examining Therapists’ Perceptions of Work-Related Stress in the late stages of the COVID-19 pandemic in Germany." The study is highly relevant and aligns well with the aims and scope of the journal. It effectively reflects the content of the manuscript, and the topic itself holds significant meaning. · The study aims to explore the work-related stress experienced by therapists during the late stages of the COVID-19 pandemic in Germany. The use of a mixed-methods approach, combining quantitative data from surveys (sample size: therapists n = 612, nurses n = 501, doctors n = 461) and qualitative data from semi-structured interviews (n = 4) and focus group discussions (n = 7), is commendable. The analysis of the data using descriptive statistics and thematic content analysis enhances the generalizability of the study findings. · I sincerely appreciate the efforts and contributions made by the authors to the scientific community. As a reader, there are certain points that should be addressed in order to gain a deeper understanding of the topic, methods, and results. Furthermore, addressing these points will contribute to the overall quality improvement of the manuscript. |
1. Over Quality of the Articles · The article is original and provides new insights to international readers. It offers an opportunity for further investigation and incorporates both theoretical and practical evidence with new findings. However, the Therapists reported similar stress-levels with other HCWs workers, with perceptions of stress differing depending on work context. Make it clear. |
2. Introduction and Research Hypotheses · The introduction section of the study is supported with relevant references and aligns with the current study. I really appreciate it; however, the introduction section needs some additional recent references that strengthen the completeness of the study. · I have question: What is the unique quality or novelty of this study? In a separate subsection with evidence, if you put the novelty of your study, it will sound more convincing and increase the originality of your study. · Two major concepts are confused and a reader I think that both are different: the stressors (8 types) and therapists’ perceptions of work-related stress. However, it is better to clarify the two distinct concepts and more focus on the study goal rather than using the two. |
Material and Methods · Interview and Focus Group Discussion are the kinds of quantitative parts which are used to strengthen or triangulate the quantitative study. My concern is do the author(s) want to identify therapists’ stressors or perception. However, there is no validation study (EFA and CFA) to strengthen the 8 stressors. I highly recommended the author(s) will add in the manuscript to increase the scientific quality of the paper> Please refer the attached references ( Hair et al., 2014; Kline, 2016). · I am happy if the researchers add or used Exploratory factor analysis after identifying the Eight stressors and make the study more valid: (1) concerns about maintaining quality of care, (2) uncertainty about the 21 future, (3) workload, (4) interactions with colleagues and patients, (5) PPE, (6) risk of infection, (7) insufficient information flow, and (8) lack of public and political recognition. In the modern scientific literature after identifying the dimensions of the constructs, Exploratory factor analysis (EFA) and confirmatory factor analysis (CFA) are highly recommended (Huang et al., 2007; Thompson, 2004; Worthington & Whittaker, 2006, Hair et.al, 2019). So, if you added the 8 stressors EFA and CFA make the study more valid and highly scientific. · I also recommended the authors apply convergent validity and discriminant validity to make the research more valid. Because the 8 types of stressors identified after the study of qualitative and quantitative data. Convergent validity is the relationship among the positive constructs. In addition, using the maximum shared variance (MSV) and the average variance extracted (AVE). The AVE values that exceed a threshold limit higher than 0.5 (AVE > 0.05) demonstrate good convergent validity. · Discriminant validity is the extent to which a construct is genuinely distinct from other constructs (Hair et al., 2019). Or When the MSV is lower than AVE are characterized by adequate discriminant validity (Hair et al., 2019). · How could you check multi-collinearity and common method biases specifically the quantitative section of the method. |
Results · The findings of the study highly focused on the qualitative section. However, I tis better to use both i.e. the quantitative study supported by qualitative data. · The 8 stressors need EFA, CFA, discriminant validity, convergent validity and, measurement invariance. For example, measurement invariance across the three groups of therapists, gender (male, female or other). I am highly recommended. |
· To recap my conclusion, this manuscript has gap in showing novelty of the study or gaps, methodological issues, and analytical problems, unable to show the new methodological issues, , |
· Overall, the manuscript provides valuable insights into the Shadow of the Pandemic: Examining Therapists’ Perceptions of Work-Related Stress in the late stages of the COVID-19 pandemic in Germany" . By addressing the suggestions and considerations mentioned above, the authors can further strengthen the manuscript and its contribution to the field.
|
· I believe it will make a valuable addition to the existing literature and spark further research on the topic. |

Author Response
Reviewer 1
General comments
I would like to begin by expressing my gratitude to the Author(s) for providing me with the opportunity to review the scientific work titled "In the Shadow of the Pandemic: Examining Therapists’ Perceptions of Work-Related Stress in the late stages of the COVID-19 pandemic in Germany." The study is highly relevant and aligns well with the aims and scope of the journal. It effectively reflects the content of the manuscript, and the topic itself holds significant meaning.
The study aims to explore the work-related stress experienced by therapists during the late stages of the COVID-19 pandemic in Germany. The use of a mixed-methods approach, combining quantitative data from surveys (sample size: therapists n = 612, nurses n = 501, doctors n = 461) and qualitative data from semi-structured interviews (n = 4) and focus group discussions (n = 7), is commendable. The analysis of the data using descriptive statistics and thematic content analysis enhances the generalizability of the study findings.
I sincerely appreciate the efforts and contributions made by the authors to the scientific community. As a reader, there are certain points that should be addressed in order to gain a deeper understanding of the topic, methods, and results. Furthermore, addressing these points will contribute to the overall quality improvement of the manuscript.
- Over Quality of the Articles
The article is original and provides new insights to international readers. It offers an opportunity for further investigation and incorporates both theoretical and practical evidence with new findings. However, the therapists reported similar stress-levels with other HCWs workers, with perceptions of stress differing depending on work context. Make it clear.
Response: We have edited the manuscript to better draw out the differences in the factors associated with stress in the different professional groups (therapists, nurses, doctors). We have now elaborated in the discussion on the point that therapists’ experiences of stress were often facility- and setting-specific. We illustrate this through specific examples of the differences and sector-specific challenges, notably of therapists working in the ambulatory (vs. the inpatient) sector.
- Introduction and Research Hypotheses
The introduction section of the study is supported with relevant references and aligns with the current study. I really appreciate it; however, the introduction section needs some additional recent references that strengthen the completeness of the study.
Response: We have added the following recent references to the introduction section, which are of relevance to the study and allow us to situate the results in the current literature on HCWs’ experiences of stress during the COVID-19 pandemic:
Carpi, M., et al. (2024). "Burnout and perceived stress among Italian physical therapists during the COVID-19 pandemic: a cross-sectional study." Psychol Health Med 29(4): 843-855.
Jow, S., et al. (2023). "Mental health impact of COVID-19 pandemic on therapists at an inpatient rehabilitation facility." Pm r 15(2): 168-175.
Oostlander, S. A., et al. (2024). "Practicing During the COVID-19 Pandemic: Experiences of Canadian Hospital-Based Occupational Therapists." Can J Occup Ther: 84174241245622.
Saragosa, M., et al. (2024). "From Crisis to Opportunity: A Qualitative Study on Rehabilitation Therapists' Experiences and Post-Pandemic Perspectives." Healthcare (Basel) 12(10).
Wójtowicz, D. and J. Kowalska (2023). "Analysis of the sense of occupational stress and burnout syndrome among physiotherapists during the COVID-19 pandemic." Sci Rep 13(1): 5743.
I have question: What is the unique quality or novelty of this study? In a separate subsection with evidence, if you put the novelty of your study, it will sound more convincing and increase the originality of your study.
Response: At the beginning of the discussion, we have now added a paragraph to elaborate on the unique quality and novelty of the study. We highlight that, to our knowledge, it is the first study which investigates the perceptions of stress of therapists working in Germany during the COVID-19 pandemic. We highlight that therapists are a profession that is under-researched, contributing to a lack of understanding of the challenges and needs of this professional group. We highlight that our study contributes to a better understanding of the situation of therapists and that it can therefore help to develop tailored support to reduce stress among this group of health care workers.
Two major concepts are confused and a reader I think that both are different: the stressors (8 types) and therapists’ perceptions of work-related stress. However, it is better to clarify the two distinct concepts and more focus on the study goal rather than using the two.
Response: We have revised the manuscript to clarify that, overall, we investigate therapists‘ experiences of work-related stress (as outlined in the title). We do this by applying a mixed-method approach: The quantitative analysis (which describes and contrasts stress-levels and factors associated with stress reported by therapists, doctors and nurses), provides a brief overview of stress-levels and the relative importance of different pre-identified factors associated with stress among different types of health care workers. The qualitative, thematic analysis offers an in-depth analysis of the factors that were raised by therapists specifically in in-depth interviews and focus group discussions as contributing to work-related stress. We have edited the manuscript to better differentiate between the quantitative and qualitative part of the analysis and hope that these edits have improved the text and clarified any confusion with regard to the above mentioned concepts.
Material and Methods
Interview and Focus Group Discussion are the kinds of quantitative parts which are used to strengthen or triangulate the quantitative study. My concern is do the author(s) want to identify therapists’ stressors or perception. However, there is no validation study (EFA and CFA) to strengthen the 8 stressors. I highly recommended the author(s) will add in the manuscript to increase the scientific quality of the paper> Please refer the attached references (Hair et al., 2014; Kline, 2016).
I am happy if the researchers add or used Exploratory factor analysis after identifying the Eight stressors and make the study more valid: (1) concerns about maintaining quality of care, (2) uncertainty about the 21 future, (3) workload, (4) interactions with colleagues and patients, (5) PPE, (6) risk of infection, (7) insufficient information flow, and (8) lack of public and political recognition. In the modern scientific literature after identifying the dimensions of the constructs, Exploratory factor analysis (EFA) and confirmatory factor analysis (CFA) are highly recommended (Huang et al., 2007; Thompson, 2004; Worthington & Whittaker, 2006, Hair et.al, 2019). So, if you added the 8 stressors EFA and CFA make the study more valid and highly scientific.
I also recommended the authors apply convergent validity and discriminant validity to make the research more valid. Because the 8 types of stressors identified after the study of qualitative and quantitative data. Convergent validity is the relationship among the positive constructs. In addition, using the maximum shared variance (MSV) and the average variance extracted (AVE). The AVE values that exceed a threshold limit higher than 0.5 (AVE > 0.05) demonstrate good convergent validity.
Discriminant validity is the extent to which a construct is genuinely distinct from other constructs (Hair et al., 2019). Or When the MSV is lower than AVE are characterized by adequate discriminant validity (Hair et al., 2019).
How could you check multi-collinearity and common method biases specifically the quantitative section of the method.
Response: We thank the reviewer for putting thought into suggesting additional methods of statistical techniques. These suggestions are based on the assumption that the eight stressors were part of the quantitative arm of the study. We understand how the original wording in the manuscript has misled the reviewer to think that the stressors were analysed quantitatively and have edited the manuscript thoroughly to explain that the study consisted of a quantitative analysis of stress as reported by therapists, nurses and doctors in a survey on the one hand, and a qualitative analysis of in-depth interviews and focus group discussions involving therapists. The two arms of the study were conducted in parallel, with data being triangulated during the analysis. The stressors which are described in detail in the findings section derive solely from the qualitative analysis. Therefore, we cannot perform the suggested statistical analyses.
Results
The findings of the study highly focused on the qualitative section. However, I tis better to use both i.e. the quantitative study supported by qualitative data.
The 8 stressors need EFA, CFA, discriminant validity, convergent validity and, measurement invariance. For example, measurement invariance across the three groups of therapists, gender (male, female or other). I am highly recommended.
Response: As stated above, we have edited the manuscript to differentiate between the quantitative and the qualitative part of the study, highlighting that these two parts were conducted in parallel. The quantitative analysis identifies the stress-levels and pre-identified factors associated with stress as reported in a survey by therapists, doctors and nurses. The paper then presents data from the qualitative, thematic analysis which drew on data provided by therapists only. By combining the quantitative and qualitative analyses, we believe that the manuscript provides comprehensive insights into therapists’ experiences of stress during the COVID-19 pandemic.
To recap my conclusion, this manuscript has gap in showing novelty of the study or gaps, methodological issues, and analytical problems, unable to show the new methodological issues, ,
Response: As outlined above, we have now added a paragraph to the discussion in which we elaborate on the unique quality and novelty of the study.
Overall, the manuscript provides valuable insights into the Shadow of the Pandemic: Examining Therapists’ Perceptions of Work-Related Stress in the late stages of the COVID-19 pandemic in Germany". By addressing the suggestions and considerations mentioned above, the authors can further strengthen the manuscript and its contribution to the field.
I believe it will make a valuable addition to the existing literature and spark further research on the topic.

Reviewer 2 Report
Comments and Suggestions for Authors
The article appears to be part of a wider and more complete study, and therefore has some limitations.
However, these limitations can be easily corrected, namely by correcting some of the content marked below and by presenting analyses to assess the statistical significance of any differences between the distribution of responses by the different categories of the criterion variables used (even though this is a qualitative study, the inclusion of this information is pertinent in order to eliminate the possibility that differences in the distribution of responses could be attributed to chance). Moreover, this information is also essential to support some of the conclusions presented by the authors in section "4. Discussion".
It is recommended to make these changes before making the decision to publish.
Specific comments:
1) The study mentions a large sample (n=1547) in the abstract (and in section 3.1. Description of the sample) and a smaller sample (lines 139-142 and Table 2). It would be important to clarify whether the study focuses on the smaller sample (n=15) or the larger sample (n=1547). Please clarify.
2) Assuming that the study focuses on a smaller sample (n=15), in addition to the percentages, it is important to include in Figure 1 the number of cases for each of the categories of answers presented. Please correct.
3) The acronyms HCW, PPE, SSI, FGD, SLT, etc. do not have plural, please correct.
4) In Table 1, the authors use the expression "not specified" to identify non-responses (missings), but in the footnote to Table 1 they already use the expression "no information", so they should standardize the use of the expression in both cases and possibly clarify whether a non-response/missing ("no information") corresponds to a "not specified" response. Please clarify.
5) Between lines 180 and 189, and in several other lines that follow, quotation marks and italics are used to mark extracts from the interviews. For the sake of parsimony, it is suggested that only one system be used: either quotation marks without italics or italics without quotation marks.
6) In line 316 of section “4. Discussion”, the authors mention "The quantitative analysis identifies…", but no quantitative analysis is presented. It is suggested that the authors use some kind of statistical test to assess the possible significance of the differences between the values of the answers given to the different categories of the criterion variables used.
The inclusion of this statistical analysis should later be used to provide empirical support for some of the conclusions the authors suggest in the following paragraphs. For example, in lines 320-321 ("Our analysis shows that for therapists, having to work with PPE and the risk of a COVID-19 infection were important sources of work-related stress."), in lines 333-334 ("Therapists' stress experiences during the pandemic differed significantly depending on the type of health facility and work setting") and in several others that follow (Lines 352-354, 367-368 ...). Please complete.
7) In the final references, the expression "et al." is used to designate different authors of several references. Please indicate the names of all the authors (and remove et al.). Please correct.
8) In the final references, the use of capital letters in the designation of the different words in the titles of journal articles should be standardized, for example, in reference [2], all the words in the title of the article begin with capital letters, as opposed to the other titles, which is repeated in several other cases ([4], [5], [10], [15]...). Please correct.
Author Response
Reviewer 2
The article appears to be part of a wider and more complete study, and therefore has some limitations.
However, these limitations can be easily corrected, namely by correcting some of the content marked below and by presenting analyses to assess the statistical significance of any differences between the distribution of responses by the different categories of the criterion variables used (even though this is a qualitative study, the inclusion of this information is pertinent in order to eliminate the possibility that differences in the distribution of responses could be attributed to chance). Moreover, this information is also essential to support some of the conclusions presented by the authors in section "4. Discussion".
It is recommended to make these changes before making the decision to publish.
Specific comments:
1) The study mentions a large sample (n=1547) in the abstract (and in section 3.1. Description of the sample) and a smaller sample (lines 139-142 and Table 2). It would be important to clarify whether the study focuses on the smaller sample (n=15) or the larger sample (n=1547). Please clarify.
Response: We have edited the manuscript to clarify the following: The statistical analysis was conducted on the large sample of study participants (n=1547), consisting of therapists, doctors and nurses, who filled in the survey. The thematic analysis was conducted on the smaller sample (n=15) of study participants (therapists only) who participated in either a 1:1 interview or a focus group discussion. The findings are presented after each other. We understand how the original wording in the manuscript has caused some confusion and has not made it clear enough that the study applied the following mixed-method approach: The study consists of a quantitative and a qualitative part and these two parts were conducted in parallel. The quantitative analysis identifies the stress-levels and pre-identified factors associated with stress as reported in a survey by therapists, doctors and nurses. The paper then presents data from the qualitative, thematic analysis which drew on data provided by therapists only. We edited the manuscript throughout to make this clearer.
2) Assuming that the study focuses on a smaller sample (n=15), in addition to the percentages, it is important to include in Figure 1 the number of cases for each of the categories of answers presented. Please correct.
Response: Given that the analysis presented in figure 1 relates to the large sample of n=1547 and in order to be in line with how such data is presented in other studies published in the same journal (e.g. https://www.mdpi.com/2227-9032/12/15/1538), we have decided to stick to percentages only.
3) The acronyms HCW, PPE, SSI, FGD, SLT, etc. do not have plural, please correct.
Response: We have corrected the acronyms and use singular only.
4) In Table 1, the authors use the expression "not specified" to identify non-responses (missings), but in the footnote to Table 1 they already use the expression "no information", so they should standardize the use of the expression in both cases and possibly clarify whether a non-response/missing ("no information") corresponds to a "not specified" response. Please clarify.
Response: As suggested, we have aligned the terminology and now only use the term „not specified“. The footnote and the term „no information“ was actually obsolete.
5) Between lines 180 and 189, and in several other lines that follow, quotation marks and italics are used to mark extracts from the interviews. For the sake of parsimony, it is suggested that only one system be used: either quotation marks without italics or italics without quotation marks.
Response: We have edited the manuscript and now stick to quotation marks to indicate extracts from the interviews and focus group discussions.
6) In line 316 of section “4. Discussion”, the authors mention "The quantitative analysis identifies…", but no quantitative analysis is presented. It is suggested that the authors use some kind of statistical test to assess the possible significance of the differences between the values of the answers given to the different categories of the criterion variables used.
The inclusion of this statistical analysis should later be used to provide empirical support for some of the conclusions the authors suggest in the following paragraphs. For example, in lines 320-321 ("Our analysis shows that for therapists, having to work with PPE and the risk of a COVID-19 infection were important sources of work-related stress."), in lines 333-334 ("Therapists' stress experiences during the pandemic differed significantly depending on the type of health facility and work setting") and in several others that follow (Lines 352-354, 367-368 ...). Please complete.
Response: We conducted a simple quantitative analysis of the stress-levels and factors associated with stress in the different professional groups, drawing on the survey data. No further statistical analysis and no significance test was performed. We have carefully edited the manuscript to not suggest that we conducted any detailed statistical analysis.
We re-worded the mentioned sentence as follows: “The quantitative as well as the qualitative analysis shows that for therapists, having to work with PPE and the risk of a COVID-19 infection were important sources of work-related stress.“ We also deleted any wording that might mislead the reader to think that we conducted significance tests.
7) In the final references, the expression "et al." is used to designate different authors of several references. Please indicate the names of all the authors (and remove et al.). Please correct.
Response: The references have been edited according to this suggestion.
8) In the final references, the use of capital letters in the designation of the different words in the titles of journal articles should be standardized, for example, in reference [2], all the words in the title of the article begin with capital letters, as opposed to the other titles, which is repeated in several other cases ([4], [5], [10], [15]...). Please correct.
Response: The references have been edited according to this suggestion.
